



# Distinct surface response to black carbon aerosols

Tao Tang[1], Drew Shindell[2], Yuqiang Zhang[2], Apostolos Voulgarakis[3,4], Jean-Francois Lamarque[5], Gunnar Myhre[6], Gregory Faluvegi[7,8], Bjørn H. Samset[6], Timothy Andrews[9], Dirk Olivié[10], Toshihiko Takemura[11], Xuhui Lee[1]

[1]School of the Environment, Yale University, New Haven, CT, USA
[2]Division of Earth and Climate Sciences, Duke University, Durham, NC, USA
[3]Leverhulmn Centre for Wildfires, Environment and Society, Department of Physics, Imperial College London, London, UK
[4]School of Environmental Engineering, Technical University of Crete, Chania, Greece.
[5]National Center for Atmospheric Research, Boulder, CO, USA
[6]CICERO, Center for International Climate and Environment Research, Oslo, Norway
[7]Center for Climate System Research, Columbia University, New York, NY, USA
[8]NASA Goddard Institute for Space Studies, New York, NY, USA
[9]Met Office Hadley Centre, Exeter, UK
[10]Norwegian Meteorological Institute, Oslo, Norway
[11]Kyushu University, Fukuoka, Japan

*Correspondence to*: Tao Tang (tao.tang@yale.edu)



**Abstract.** For the radiative impact of individual climate forcings, most previous studies focused on the global mean values at

the top of the atmosphere (TOA) and less attention has been paid to surface processes, especially for black carbon aerosols. In

this study, the surface radiative responses to five different forcing agents were analyzed by using idealized model simulations.

Our analyses reveal that for greenhouse gases, solar irradiance and scattering aerosols, the surface temperature changes are

mainly dictated by the changes of surface radiative heating, but for BC, surface energy redistribution between different

components plays a more crucial role. Globally, when a unit BC forcing was imposed at TOA, the net shortwave radiation at

the surface decreased by $5.09\pm1.80$ W m$^{-2}$ (averaged over global land), which is partially offset by increased downward

longwave radiation ($1.67\pm0.24$ W m$^{-2}$) from the warmer atmosphere, causing a net decrease in the incoming downward surface

radiation of $3.42\pm0.51$ W m$^{-2}$. Despite a reduction in the downward radiation energy, the surface air temperature still increased

by $0.14\pm0.05$ K because of less efficient energy dissipation, manifested by reduced surface sensible ($2.53\pm0.37$ W m$^{-2}$) and

latent heat flux ($1.30\pm0.27$ W m$^{-2}$), as well as a decrease of Bowen ratio ($0.18\pm0.05$). Such reductions of turbulent fluxes can

be largely explained by enhanced air stability ($0.06\pm0.01$ K), measured as the difference of the potential temperature between

925 hPa and surface, and reduced surface wind speed ($0.05\pm0.01$ m s$^{-1}$). The enhanced stability is due to the faster atmospheric

warming relative to the surface whereas the reduced wind speed can be partially explained by enhanced stability and reduced

equator-to-pole atmospheric temperature gradient. These rapid adjustments under BC forcing exerted a 'top-down' impact on

the surface energy redistribution and thus, surface temperature response, which is not observed under greenhouse gases or

scattering aerosols. Our study provides new insights into the impact of absorbing aerosols on surface energy balance and

surface temperature response.

## 1 Introduction

Black carbon (BC) aerosols, emitted from diesel engines, biofuels, forest fires incomplete combustion and biomass burning,

could significantly impact the Earth's climate by changing its radiative balance or by perturbing the hydrological cycle

(Ramanathan et al., 2001; Menon et al., 2002). The former is realized via absorbing solar radiation, causing positive effective

radiative forcing (ERF) at the top of the atmosphere (TOA) and thus, warming the climate (Ramanathan & Carmichael, 2008;

Bond et al., 2013; Myhre et al., 2013b) while the latter is partly through modifying the microphysical properties of clouds

(e.g., albedo and lifetime) (Koch & Del Genio, 2010; Bond et al., 2013; Boucher et al., 2013), which could further impact

ERF. For instance, Menon et al. (2002) attributed the cooling and drying trends in North China in the second half of 20[th]

century to BC aerosols; Meehl et al. (2008) suggested that BC contributed to the precipitation change in India by altering the

meridional temperature gradient.

For radiative impacts, however, most previous studies have only focused on TOA forcing. TOA forcing is useful in

understanding the climate feedback, climate sensitivity, and future climate change (Andrews et al., 2012), but it is not

necessarily predictive of the spatial pattern of surface temperature response, which is more related to surface radiative changes





(Wild et al., 2004). One intriguing phenomenon for BC is that, on the global scale, BC could warm the surface even with reduced solar radiation and net radiation at the surface (Ramanathan & Carmichael, 2008), which is somewhat counterintuitive as higher surface temperature generally requires more incoming radiation at the surface. FAQ 7.2 of Boucher et al. (2013) briefly described the heating process induced by BC. Specifically, BC particles firstly heat the atmosphere and cause surface

cooling locally, but then warm both the surface and the atmosphere due to atmospheric circulation and mixing processes. When it comes to surface response, Ramanathan et al. (2001) suggested that the reduced solar radiation at the surface is possibly counteracted by reduced evaporation, which further perturbs the hydrological cycle. Krishnan and Ramanathan (2002) found that the source regions of haze are subject to cooling due to the absorption of solar radiation whereas regions outside the source can be warming, thus contributing to overall global warming. Jacobson and Kaufman (2006) noted that aerosols can reduce

wind speed in some coastal regions and also reduce evaporation through their enhancement of clouds. Wilcox et al. (2016) reported a reduction of turbulent flux under BC aerosols at the surface and linked such responses to clouds. Based on model simulations, Myhre et al. (2018) concluded that BC aerosols can change the global hydrological cycle by suppressing sensible heat flux at the surface, and attributed this suppression to the changes of air stability.

The published studies citied above provide informative insights to the surface radiative responses to BC aerosols. To our best knowledge, however, a clear and detailed mechanism of surface radiative response to BC is still lacking from the perspective of the surface energy balance. In this study, we aim to fill this gap and answer the following scientific questions: (i) How does the surface warming under BC aerosols differ from warming due to greenhouse gases and solar forcing; (ii) What are the specific mechanisms that drive such warming responses; and (iii) what are the relative contribution to the surface temperature

change from each surface energy budget component.

## 2 Data and Methods

### 2.1 Data

This study employs the model output from the Precipitation Driver and Response Model Intercomparison Project (PDRMIP), utilizing simulations examining the climate responses to individual climate drivers (Myhre et al., 2017). The eight models used

in this study are CanESM2, GISS-E2R, HadGEM2-ES, HadGEM3, MIROC, CESM-CAM4, CESM-CAM5 and NorESM. The versions of these models are essentially the same as their versions in the $5^{th}$ Assessment Report of Intergovernmental Panel on Climate Change (IPCC AR5). The configurations and basic settings are listed in Table 1. In these simulations, five separate perturbations were applied to all the models instantly on global scale: a doubling of $CO_2$ concentration ($CO_2 \times 2$), a tripling of $CH_4$ concentration ($CH_4 \times 3$), a 2% increase in solar irradiance (Solar+2%), a tenfold increase of present-day black carbon

concentration/emission ($BC \times 10$), and a fivefold increase of present-day $SO_4$ concentration/emission ($SO_4 \times 5$). Each perturbation was run in two parallel configurations, a 15-year fixed sea surface temperature (fsst) simulation and a 100-year coupled simulation. The former is compared with its fsst control simulation to diagnose the ERF and fast responses in each




model, whereas the latter is used to examine climate responses. One model (CESM-CAM4) used a slab ocean setup for the coupled simulation whereas the others used a full dynamic ocean. For aerosol perturbations, monthly year 2000 concentrations

were derived from the AeroCom Phase II initiative (Myhre et al., 2013a) and multiplied by the stated factors in concentration-driven models. Some models were unable to perform simulations with prescribed concentrations. These models multiplied emissions by these factors instead (Table 1). The aerosol loadings in the NorESM model for the two aerosol perturbations are shown in Fig. 1 for an illustrative purpose; the spatial patterns are similar for other models. In the BC experiment, the concentration is highest in East China (E. China), followed by India, tropical Africa and South America (S. America). In the

current study, these four regions are referred to as source regions due to their high emissions while US and Europe are defined as non-source regions due to their relative low emissions. For the $SO_4$ experiment, the aerosols are mainly restricted to the Northern Hemisphere (NH), with the highest loading observed in E. China, followed by India and Europe. The eastern US also has moderately high concentrations. More detailed descriptions of PDRMIP and some PDRMIP findings are given in Samset et al. (2016), Myhre et al. (2017) and Tang et al. (2018).

**2.2 Methods**

In this study, we start from the surface energy balance, and restrict our discussions to land grids only. The incoming radiative energy ($R_{in}$) includes:

$$R_{in} = {\downarrow}SW - {\uparrow}SW + {\downarrow}LW \qquad (1)$$


In Eqn. (1), ${\downarrow}SW$ represents downward shortwave radiation and ${\uparrow}SW$ represents reflected SW radiation. ${\downarrow}LW$ denotes the downward LW radiation. The law of energy conservation requires that the $R_{in}$ should be balanced by the outgoing energy ($E_{out}$):

$$E_{out} = {\uparrow}LW + H + \lambda E + G \qquad (2)$$

In Eqn (2), ${\uparrow}LW$ is the outgoing longwave radiation, which is a function of temperature based on the Stefan-Boltzmann law. H, $\lambda E$ and G denote sensible heat flux, latent heat flux and ground heat flux, respectively. For latent heat flux ($\lambda E$), $\lambda$ is the specific latent heat of evaporation and E is the evaporation rate. $R_{in}$ is defined as surface radiative heating, as it is the radiative

input provided to the surface to raise the surface temperature (Wild et al., 2004). The surface responds to the imposed energy by redistributing the energy content through each $E_{out}$ component. Since $R_{in}$ is equal to $E_{out}$, we have:

$$\triangle R_{in} = \triangle {\uparrow}LW + \triangle H + \triangle \lambda E + \triangle G \qquad (3)$$



The changes of each energy component, denoted by △, are obtained by subtracting the control simulations from the perturbations using the data of years 6-15 in each fsst simulation and years 71-100 in each coupled simulation. The changes are then normalized by the ERF in the corresponding experiments to obtain the changes per unit global forcing. The ERF values for each model are obtained from Tang et al. (2019), which diagnosed as the combination of net SW radiation plus the downward LW radiation at the TOA in the fsst simulations (Hansen et al., 2002). The multi-model mean (MMM) ERF values

are $3.68\pm0.09$ W m$^{-2}$ (CO$_2\times$2), $1.15\pm0.09$ W m$^{-2}$ (CH$_4\times$3), $4.21\pm0.05$ W m$^{-2}$ (Solar+2%), $1.20\pm0.28$ W m$^{-2}$ (BC$\times$10), and -$3.63\pm0.71$ W m$^{-2}$ (SO$_4\times$5) for indicated experiments (mean $\pm$ 1 standard error). The MMM changes are estimated by averaging all the eight models' results. A two-sided student t-test is used to examine whether the MMM results are significantly different from zero. The same process was repeated for all variables analyzed in the current study.

**3 Results**

**3.1 Incoming radiation and surface temperature changes under BC forcing**

Figure 2a-c show the MMM changes of $R_{in}$ and its components for the fsst simulations of the BC experiment. The fsst simulations were analyzed because we mainly focus on the rapid adjustments when the forcing is instantly imposed. Rapid adjustments are generally referred to the fast responses that affect the components of the climate system and modify the global energy budget indirectly. Unlike feedbacks, rapid adjustments do not operate through changes in the global mean temperature

and most are thought to occur within a few weeks (Boucher et al., 2013). Specifically, when a unit BC forcing was imposed at the TOA, the net surface SW radiation decreased by $5.09\pm1.80$ W m$^{-2}$ due to the absorption of solar radiation by BC particles whereas ↓LW radiation shows an increase of $1.67\pm0.24$ W m$^{-2}$ (Fig. 2a & b), as a result of the warmer atmosphere. When combined, $R_{in}$ still decreased by $3.42\pm0.51$ W m$^{-2}$ on the global scale, with some positive changes only in high-latitude regions (Fig. 2c). However, the surface air temperature increased globally by $\triangle T = 0.14 \pm 0.05$ K despite the decreased $R_{in}$, except in

the source regions where some slight cooling trends occurred (Fig. 2d). It is noted that these results are for land grids only. The pattern of cooling in the source regions and warming elsewhere agrees well with findings reported by Krishnan and Ramanathan (2002). This type of changes persisted into near-equilibrium state, where global mean temperature changed and associated feedbacks were included (Fig. 2e-h). Due to the enhanced warming of the atmosphere and probably water vapor buildup, the ↓LW radiation shows a stronger increase (Fig. 2f), making the $R_{in}$ mostly positive and therefore positive

temperature change, except for source regions (Fig. 2g & h). An open question is how the temperature increased with a decreasing $R_{in}$ in the rapid adjustment processes. In order to better understand the mechanisms behind this warming phenomenon, we will explore the $E_{out}$ components in the next section.



### 3.2 Decomposition of outgoing energy

Figure 3 depicts the MMM changes of $R_{in}$ and all components of $E_{out}$ in the fsst simulations for all five experiments. The spatial

patterns of $\triangle R_{in}$ and $\triangle \uparrow LW$ for the BC experiment were quite different (Fig. 3d & i). Another notable feature is the significant reductions of sensible and latent heat flux in the BC experiment (Fig. 3n & s), which is in agreement with previous studies (Wilcox et al., 2016; Myhre et al., 2018; Suzuki & Takemura, 2019).

These changes are obvious when averaged globally (Fig. 4a). The $R_{in}$ decreased by 3.42±0.51 W m$^{-2}$, and the H and λE

decreased by 2.53±0.37 W m$^{-2}$ and 1.30±0.27 W m$^{-2}$, respectively, making the energy partitioned to $\triangle \uparrow LW$ positive (0.37±0.19 W m$^{-2}$). In other words, although the radiative heating ($R_{in}$) decreased, convective and evaporative cooling decreased by a larger amount (112% relative to $\triangle R_{in}$) owing to less efficient energy dissipation, thereby warming the surface and a positive $\triangle \uparrow LW$ radiation. The reduction of turbulent fluxes (H and λE) is found for both source regions and non-source regions (Fig. 4b). In the source regions, the reductions of turbulent flux were nearly the same as the reduction of $R_{in}$, making

the temperature response negligible or only slightly negative. In the non-source regions, the reduction of turbulent fluxes exceeded the reduction of $R_{in}$, making the temperature response positive. Another interesting phenomenon is that the reduction of H is larger than the reduction of λE for the BC experiment, both on the global scale and in the source regions (Fig. 4). The larger reduction of H indicates a decrease of Bowen ratio (β), defined as the ratio of sensible heat flux over latent heat flux. Globally, β decreased by 0.18±0.05 in the BC case, and such drop could reach 0.35 ± 0.15 in the source regions. In comparison,

under other forcing agents, the changes of β were much smaller (the global MMM changes within 0.03).

Under other forcing agents, the spatial patterns of $\triangle \uparrow LW$ are similar to the patterns of $\triangle R_{in}$. The changes of H and λE were relatively small and sometimes canceled out each other, making little contributions to temperature change compared with BC (Fig. 3 and 4a). Therefore, the temperature change ($\triangle \uparrow LW$) was dominated by the change of radiative heating ($\triangle R_{in}$). It is

worth noting that the decrease of λE in the $CO_2$ experiment is due the physiological effect of vegetation and plantation (Fig. 3p), which is included in the PDRMIP models (Richardson et al., 2018). The contributions of G are negligible (Fig. 3u-y) and will not be further discussed.

### 3.3 Attribution of temperature change

In order to quantify the contributions of each component to $\triangle T$, we applied a multi-linear regression model to the MMM

values of $\triangle T$ and the energy components in each experiment, as $\triangle T = a \times \triangle R_{in} + b \times \triangle H + c \times \triangle \lambda E$. Here $\triangle R_{in}$ represents the changes of radiative heating and $\triangle H$ and $\triangle \lambda E$ denote changes in the surface energy redistribution and all grid were given equal weight. The results are listed in Table 2. A point-wise comparison of original $\triangle T$ and fitted $\triangle T$ is shown in Fig. S1. These regressions reproduced the $\triangle T$ fairly well, since the correlation coefficients between $\triangle T$ and fitted $\triangle T$ are all above





0.73 and most of the data points align along the one-one line. For $CO_2$, $CH_4$, Solar and sulphate aerosols, the coefficients of

$\triangle R_{in}$ are one order of magnitude larger than the coefficients of the turbulent fluxes, suggesting that $\triangle R_{in}$ dominates the

temperature change under these forcing agents. When it comes to BC, however, $\triangle T$ is more sensitive to $\triangle H$, followed by $\triangle R_{in}$ and $\triangle \lambda E$. With the regression coefficients, we estimated the contributions of each energy component to $\triangle T$ (Fig. 5). In

line with our previous results, $\triangle T$ was dominated by surface heating ($\triangle R_{in}$) for most forcing agents with very limited role

from turbulent fluxes. BC, nonetheless, is an exception. For BC aerosols, $\triangle T$ was influenced by both surface heating and

turbulent fluxes, with the cooling from the former being overwhelmed by the warming from the latter (Fig. 5n, s and x). The

domain-averaged changes for the BC experiment are listed in Table 3. Globally, $\triangle H$ produced 0.54 K warming and $\triangle \lambda E$ led

to 0.20 K warming. The combined 0.74 K warming was offset by 0.60 K cooling attributed to reduced $\triangle R_{in}$, producing a net

warming of 0.14 K. In terms of percentage, $\triangle H$ and $\triangle \lambda E$ contributed 73% and 27% respectively to the total warming. Such

patterns are also seen on regional scales. The warming contributions from $\triangle H$ were 37% (US), 49% (Europe), 76% (E. China),

84% (India), 69% (Africa) and 81% (S. America) and the remaining part were contributed by $\triangle \lambda E$.

### 3.4 Mechanisms underlying the reduction of turbulent fluxes

The above analyses show that for most of the forcing agents, $\triangle R_{in}$ dominates the surface temperature response while for BC,

the surface energy redistribution also comes into play in modifying temperature response as a result of the significant

reductions of turbulent fluxes. The next question is why turbulent fluxes decreased substantially in response to BC particles.

According to the bulk parameterization of turbulent fluxes, the sensible heat flux is expressed as $Q_{SH} = \rho C_p C_H U(T_s - T_a)$ and

latent heat flux as $Q_{LH} = \rho L_v C_E U(q_s - q_a)$. In these two equations, $\rho$ is air density, $C_p$ and $L_v$ are air specific heat capacity and

latent heat of vaporization, respectively, $C_H$ and $C_E$ are two exchange coefficients, U denotes surface wind speed, and ($T_s - T_a$)

and ($q_s - q_a$) represent temperature gradient and humidity gradient between the surface and air, respectively. In the rapid

adjustment stage, wind speed (U) and temperature gradient are the two possible causes for the changes of sensible heat and

wind speed (U) is likely the only main factor driving the change of latent heat flux.

Figure 6a-e show the MMM changes of lower tropospheric stability (LTS), defined as the potential temperature difference

between 925 hPa and the surface. An enhanced stability is observed for the BC experiment; ΔLTS is 0.06±0.01 K averaged

globally, in contrast to near zero values from other experiments (0.005K to 0.016 K). The ΔLTS for the NH are even larger,

with 0.08±0.01 K for BC and 0.005 to 0.020K for other forcing agents. The enhanced LTS, which can significantly impact the

sensible heat flux (Myhre et al., 2018), arises from the fact that the BC layers warm faster relative to the surface due to BC

absorption of solar radiation. The changes of LTS patterns are similar for 850 hPa and 700 hPa (Fig. S2).



Figure 6f-j portrait the MMM changes of surface wind speed. BC caused a much larger decrease of wind speed with respect
to other forcing agents, 0.05±0.01 m s$^{-1}$ globally compared with zero from other forcing agents. The reduction in wind speed
explains the weakening in both sensible and latent heat fluxes according to the bulk parameterization.

Now the last question is why the surface wind speed decreased under BC forcing. The first potential explanation is the
abovementioned enhanced LTS. Jacobson and Kaufman (2006) has clearly demonstrated that the enhanced LTS and reduced
turbulent exchange can reduce the turbulent kinetic energy and vertical transport of horizonal momentum, thereby reducing
surface wind speed. The second possible explanation is that from the dynamical perspective, wind speed is controlled by the
pressure gradient force (PGF), the Coriolis force, the gravitational force and the frictional force. PGF is the driving force for
atmospheric motion and is potentially the main driver for the changes of wind speed in the current idealized experiments. On
global scale, the excessive heating in the tropics with respect to middle and high latitudes causes PGF to point toward polar
regions. Here we hypothesize that the decrease of temperature gradient between the equator and poles under BC forcing
weakened the PGF and slowed down the wind speed. Evidence in support of this hypothesis is found in Figure 7a-e showing
the zonal mean atmospheric temperature change. Mechanistically, BC caused a larger atmospheric heating in the middle
latitudes of NH (30°N~60°N) relative to tropics due to more of the BC forcing being located at middle latitudes of NH (Fig.
7d). The faster warming of middle latitudes weakened the PGF between the equator and polar regions, as seen from the larger
increases of geopotential height of 500 hPa in the middle latitude regions (Fig. 7i). These patterns are not observed in the other
experiments. The changes of geopotential height at other levels show similar results (Fig. S3).

To further understand the relationship between changes in wind speed and temperature, we defined a temperature gradient
index (Allen et al., 2012) as $2 \times \triangle T_{30-60} - (\triangle T_{0-30} + \triangle T_{60-90})$, where $\triangle T$ is the mass-weighted (300~850 hPa) temperature
response, and subscripts 0-30, 30-60 and 60-90 denote low (0°~30°N), middle (30°N~60°N), and high (60°N~90°N) latitudinal
zones, respectively. When the index becomes more positive, middle latitudes warm faster, and a stronger reduction of wind
speed is expected. The results for each individual model and experiment are shown in Fig. 8. A reasonably good correlation is
seen in the BC scenario (r = -0.59): a larger change in the temperature gradient index corresponds to a stronger decrease in
wind speed. The results for other experiments are mostly scattered around zero.


On regional and local scales, several other factors might also contribute to surface wind change (Wu et al., 2018). For instance,
the aerosols in Asia have been reported to modify the land-sea temperature contrast, and thus modify monsoon circulation (Xu
et al., 2006; Meehl et al., 2008). The different phases of internal variability (e.g., ENSO and NAO) could modulate the
circulations on interannual to multi-decadal time scales (Jerez et al., 2013; Hu & Fedorov, 2018). Bichet et al. (2012) suggested
that changes in the surface roughness length may also change the wind speed. These factors are not considered in the present
study.





## 4 Discussion and Summary

Our analyses demonstrate that under BC forcing, surface energy redistribution plays a vital role in modifying the surface temperature due to the changes in turbulent fluxes. The changes of turbulent fluxes are consequences of a 'top-down' influence.

The warming BC layers resulting from absorption of solar radiation enhances air stability and reduces wind speed. As a result, the surface turbulent fluxes are suppressed. This mechanism is not observed for other forcing agents such as greenhouse gases and scattering aerosols. A similar 'top-down' mechanism has been observed in the solar forcing, in which the stratosphere ozone reacts to the UV part of the solar variability and produces additional heating, leading to changes of circulation in the stratosphere. The changes in the stratosphere modify tropical tropospheric circulation that may impact the surface climate

(Haigh, 1996). It is noted that the 'top-down' mechanism of solar forcing is not included in the current PDRMIP models. The solar experiments in our analyses are restricted to the surface response to increased solar radiation, which is 'bottom-up' mechanism.

As noted in section 3.1, our above analyses mainly focus on the rapid adjustments, which are part of ERF by definition

(Boucher et al., 2013). For the BC experiment, these adjustments drive the surface to respond to the forcing. Most of the changes seen in the rapid adjustment stage extended into the near-equilibrium (Fig. S4-S5). For BC forcing, $\triangle R_{in}$ in near-equilibrium state is close to zero with large inter-model spread ($-0.34\pm1.12$ W m$^{-2}$) instead of strong negative values ($-3.42\pm0.51$ W m$^{-2}$) in the rapid adjustment stage. The near-zero $R_{in}$ in equilibrium is mainly due to larger ↓LW radiation and feedbacks. The equilibrium turbulent flux H and $\lambda$E are lowered by $2.15\pm0.36$ W m$^{-2}$ and $1.45\pm0.29$ W m$^{-2}$ respectively, which

are comparable in magnitude to changes in the rapid adjustment stage. Such reductions of the equilibrium turbulent fluxes are found in both source regions and non-source regions. Since less energy dissipated away from the surface, more energy ($3.28\pm0.65$ W m$^{-2}$) was partitioned into $\triangle$LW, warming the surface by $0.49\pm0.10$ K. Fig. S6 shows the slow responses under each forcing, which are obtained by subtracting the rapid adjustments from the coupled simulations. The slow responses are driven by global mean temperature change alone. Interestingly, the spatial patterns are quite similar across different forcing

agents. It is further confirmed our finding that it is the rapid adjustment that led to the different surface responses to BC.

Two limitations exist in our current study. First, the aerosol-cloud interactions could not be fully represented, because for the models with fixed aerosol concentration, the changes of cloud lifetime do not affect aerosols. Second, for the BC simulations, two models (MIROC and NorESM) include aerosol indirect effects while the remaining ones have only aerosol-radiation

interactions included (instantaneous and rapid adjustments). The cloud effects in these two models may slightly modify the SW radiation at the surface (Tang et al., 2020), although the results from these two models do not differ qualitatively from the other models without those effects. We suggest that our conclusions are not sensitive to such cloud effects.



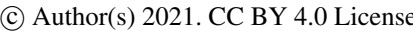

In summary, our study shows that for forcing agents such as GHG, solar and scattering aerosol, $\triangle R_{in}$ dominates the surface
temperature response. For BC forcing, the surface energy redistribution also plays an important role. Under BC forcing, the
energy is dissipated less efficiently from the surface to the lower atmosphere, which causes warming at the surface despite the
reduced radiative heating. The reductions of sensible heat flux accounts for 73% of the surface warming on global scale and
40~80% of the warming on regional scales, with the remaining part arising from the reductions of latent heat flux. Such
reductions of turbulent fluxes can be explained by enhanced lower tropospheric stability and reduced surface wind speed. The
former is attributed to a faster atmospheric warming relative to the surface whereas the latter is associated with enhanced
stability and reduced equator-to-pole atmospheric temperature gradient. These analyses contribute to our understanding of the
impact of absorbing aerosols on surface radiation and climate.

**Data code availability**

The PDRMIP model output used in this study are available to public through the Norwegian FEIDE data storage facility. For
more information, please see http://cicero.uio.no/en/PDRMIP. This study is performed by using Matlab R2019a. The Matlab
code is available upon request.

**Competing interests**

The authors declare no competing interests.

**Author contributions**

T.T. designed this study. T.T. performed data analysis and wrote the initial manuscript. All authors contributed to scientific
discussion, results framing and manuscript polishing.

**Acknowledgement**

We acknowledge the NASA High-End Computing Program through the NASA Center for Climate Simulation at Goddard
Space Flight Center for computational resources to run the GISS-E2R model and support from NASA GISS. PDRMIP is partly
funded through the Norwegian Research Council project NAPEX (project number 229778). AV is partially funded by the
Leverhulme Trust, grant RC-2018-023. Computing resources for CESM1-CAM5 (ark:/85065/d7wd3xhc) simulations were
provided by the Climate Simulation Laboratory at NCAR Computational and Information System Laboratory, sponsored by
the National Science Foundation and other agencies. X. L. acknowledges support from the US National Science Foundation
(grant AGS1933630). T. Takemura is supported by the Japan Society for the Promotion of Science (JSPS) KAKENHI (grant
no. JP19H05669), the Environment Research and Technology Development Fund (S-20) of the Environmental Restoration



and Conservation Agency, Japan, and the NEC SX supercomputer system of the National Institute for Environmental Studies, Japan. T. Andrews was supported by the Met Office Hadley Centre Climate Programme funded by BEIS and Defra and the Newton Fund through the Met Office Climate Science for Service Partnership Brazil (CSSP Brazil).









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



**Table 1.** Descriptions of the eight PDRMIP models used in this study.

| Model name | Version | Resolution | Ocean setup | Aerosol setup | references |
|---|---|---|---|---|---|
| CanESM2 | 2010 | 2.8×2.8 35 levels | Coupled | Emission | Arora et al. (2011) |
| GISS-E2R | E2-R | 2×2.5 40 levels | Coupled | Fixed concentration | Schmidt et al. (2014) |
| HadGEM2-ES | 6.6.3 | 1.875×1.25 38 levels | Coupled | Emissions | Collins et al. (2011) |
| HadGEM3-GA | 4.0 | 1.875×1.25 85 levels | Coupled | Fixed concentration | Bellouin et al. (2011) Walters et al. (2014) |
| MIROC-SPRINTARS | 5.9.0 | T85 40 levels | Coupled | HTAP2 emissions | Takemura et al. (2009) Takemura et al. (2005) Watanabe et al. (2010) |
| CESM-CAM4 | 1.0.3 | 2.5×1.9 26 levels | Slab | Fixed concentration | Neale et al. (2010) Gent et al. (2011) |
| CESM-CAM5 | 1.1.2 | 2.5×1.9 30 levels | Coupled | Emissions | Hurrell et al. (2013) Kay et al. (2015) Otto-Bliesner et al. (2016) |
| NorESM | 1-M | 2.5×1.9 26 levels | Coupled | Fixed concentration | Bentsen et al. (2013) Iversen et al. (2013) Kirkevåg et al. (2013) |

Note: HTAP2 = Hemispheric Transport Air Pollution, Phase 2.








**Table 2.** Multi-linear regression model for each experiment.

| Experiment | Regression model | Correlation coefficient (r) |
|:---:|:---:|:---:|
| $CO_2$ | $\triangle T = 0.148 \times \triangle Rin + 0.004 \times \triangle H + 0.002 \times \triangle \lambda E$ | 0.89 |
| $CH_4$ | $\triangle T = 0.143 \times \triangle Rin - 0.032 \times \triangle H - 0.030 \times \triangle \lambda E$ | 0.77 |
| Solar | $\triangle T = 0.083 \times \triangle Rin + 0.006 \times \triangle H - 0.009 \times \triangle \lambda E$ | 0.73 |
| BC | $\triangle T = 0.176 \times \triangle Rin - 0.214 \times \triangle H - 0.156 \times \triangle \lambda E$ | 0.89 |
| $SO_4$ | $\triangle T = 0.072 \times \triangle Rin + 0.009 \times \triangle H - 0.020 \times \triangle \lambda E$ | 0.77 |










**Table 3.** Domain-averaged $\triangle T$ and contributions from each radiative component estimated from the linear regression model for the BC experiment (unit: K).

| Region | $\triangle T$ | Fitted $\triangle T$ | $\triangle R_{in}$ | $\triangle H$ | $\triangle \lambda E$ |
|---|---|---|---|---|---|
| Global | 0.14 ± 0.05 | 0.14 | -0.60 | 0.54 | 0.20 |
| US | 0.53 ± 0.16 | 0.49 | -0.34 | 0.31 | 0.52 |
| Europe | 0.39 ± 0.10 | 0.33 | -0.53 | 0.42 | 0.44 |
| E. China | 0.06 ± 0.20 | -0.01 | -3.03 | 2.28 | 0.73 |
| India | 0.01 ± 0.08 | -0.06 | -2.68 | 2.20 | 0.42 |
| Africa | -0.12 ± 0.06 | -0.10 | -2.29 | 1.51 | 0.69 |
| S. America | 0.01 ± 0.05 | 0.02 | -0.91 | 0.76 | 0.18 |








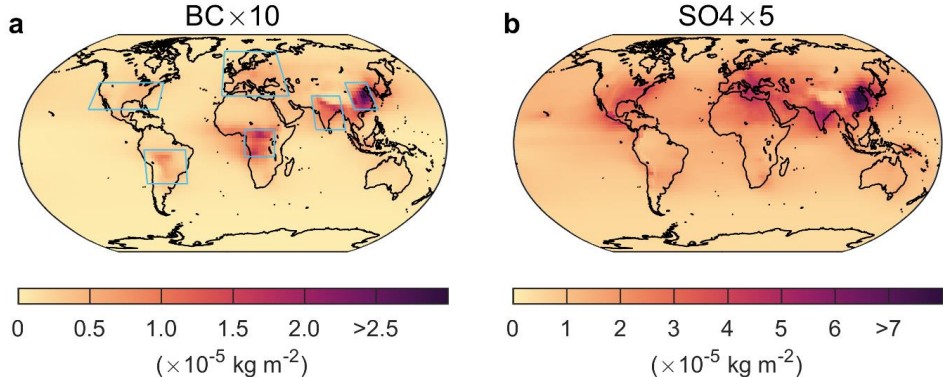

**Figure 1: Aerosol loadings for the two aerosol experiments in the NorESM model.**







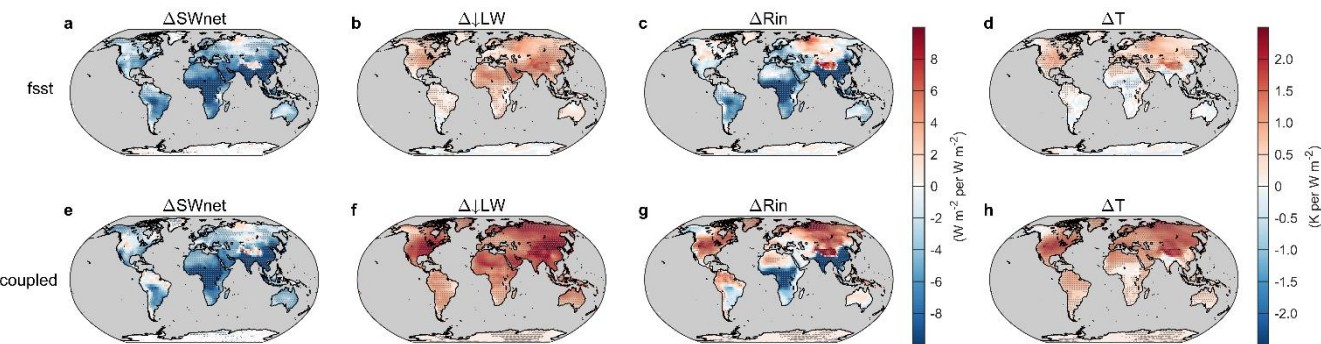

**Figure 2: MMM changes of surface net SW radiation, downward LW radiation, incoming radiation ($R_{in}$), and surface air temperature in the fsst (a-d) and coupled simulations (e-h) for the BC experiment. All changes are normalized to changes per unit global forcing. Grey dots indicate that the MMM changes are significant at a *p* value of 0.05.**








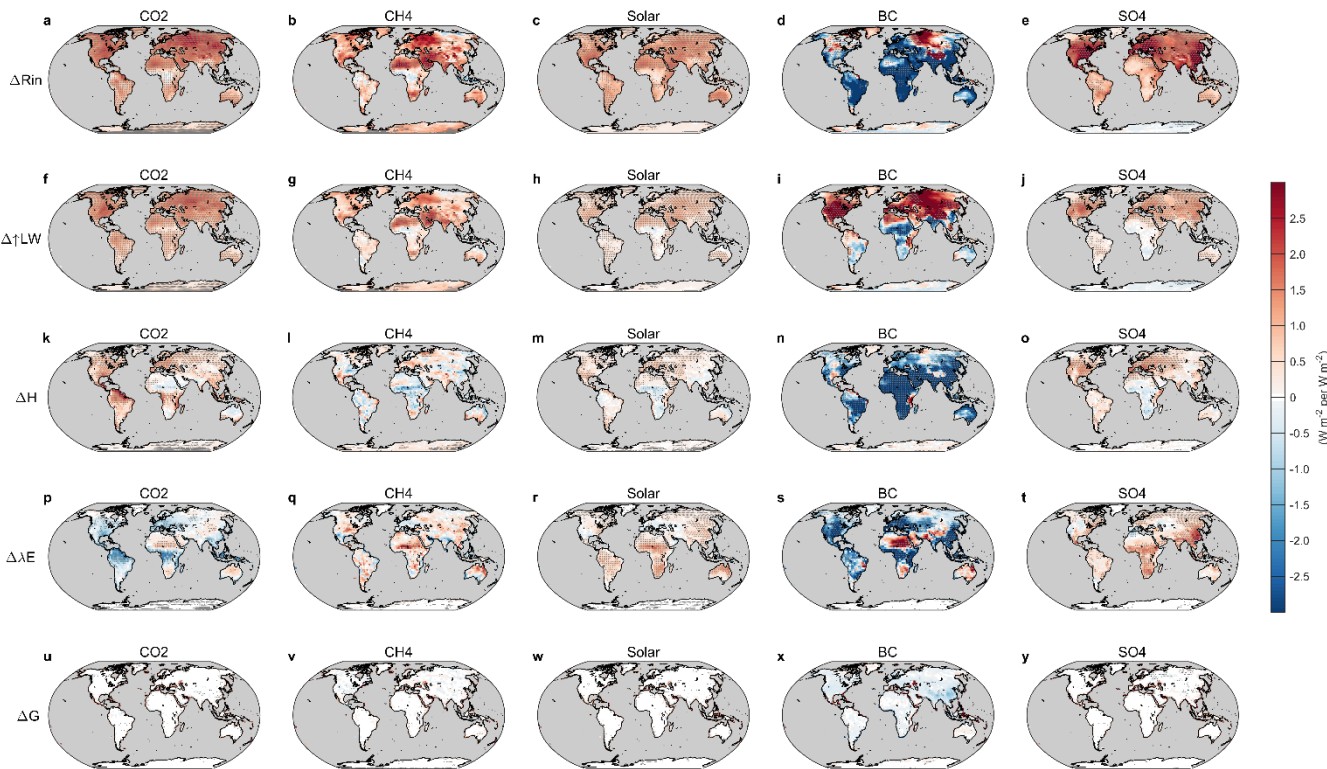

**Figure 3: MMM changes of $R_{in}$ and outgoing energy components for all five experiments in the fsst simulations. All changes are normalized to changes per unit global forcing. Grey dots indicate that the MMM changes are significant at a $p$ value of 0.05.**





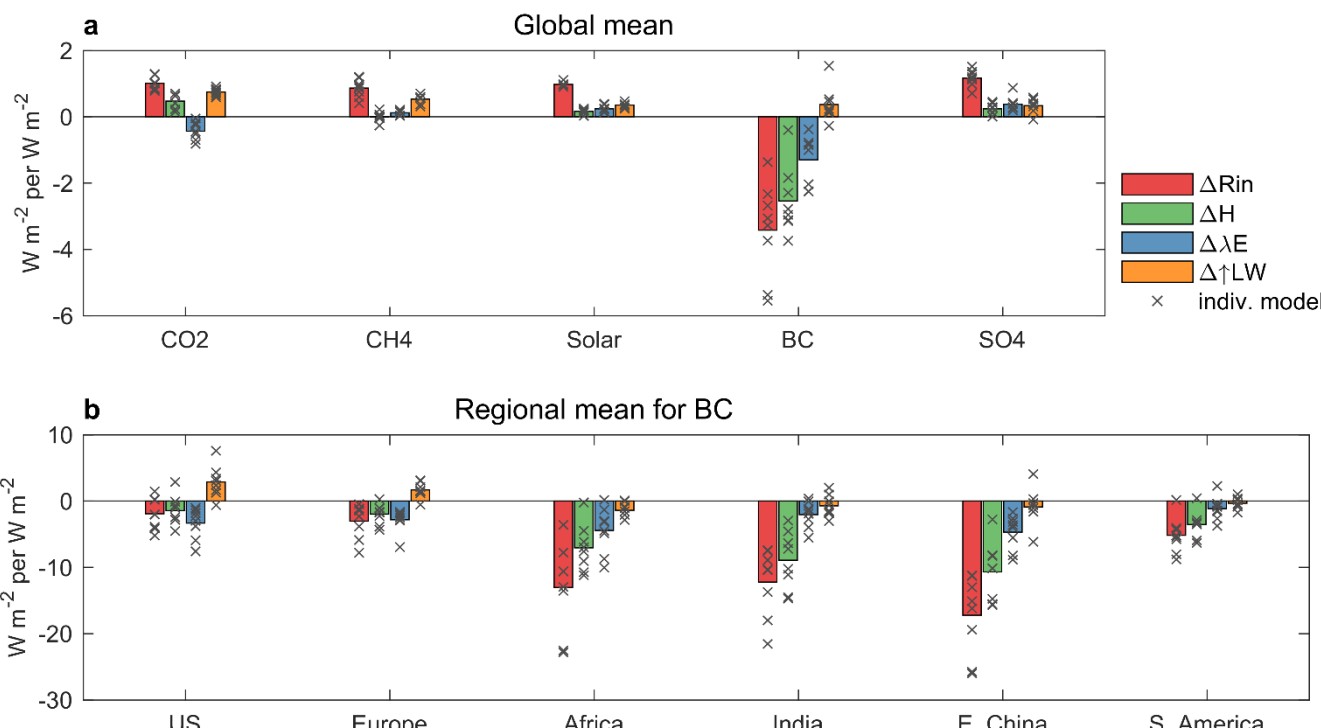

**Figure 4: Domain-averaged values of $R_{in}$ and the components of $E_{out}$ from the fsst simulations for global mean (a) and for selected regions under BC forcing (b). ✕ indicates the values from individual models.**





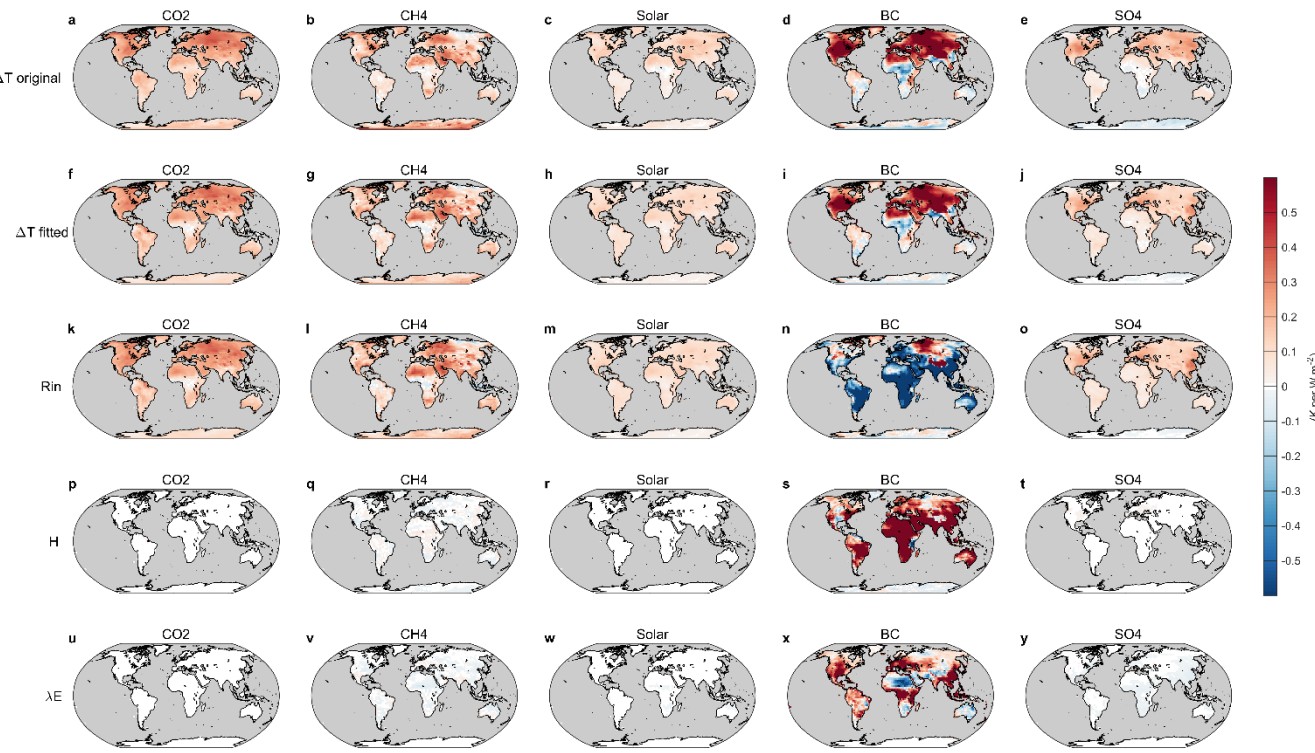

**Figure 5: Air temperature change per unit forcing. Original △T (a-e), △T estimated from multi-linear regression model (f-j), and temperature change contributed by each component based on the linear regression models (k-y).**

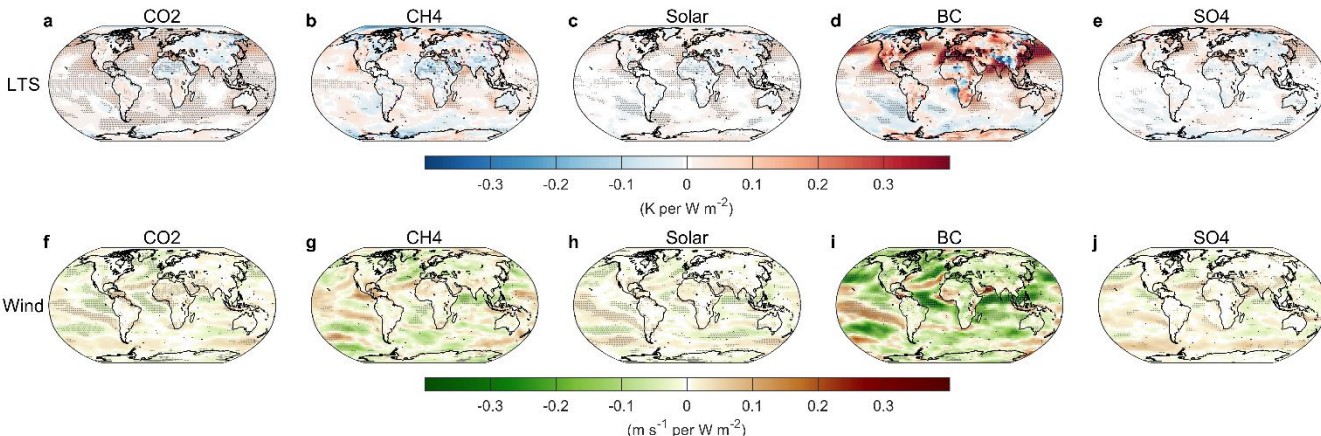

**Figure 6: MMM changes for lower tropospheric stability (LTS, a-e) and surface wind velocity (f-j) per unit global TOA forcing. For LTS, positive anomalies indicate a more stable atmosphere. Grey dots indicate that the MMM changes are significant at a $p$ value of 0.05.**

630

635

640



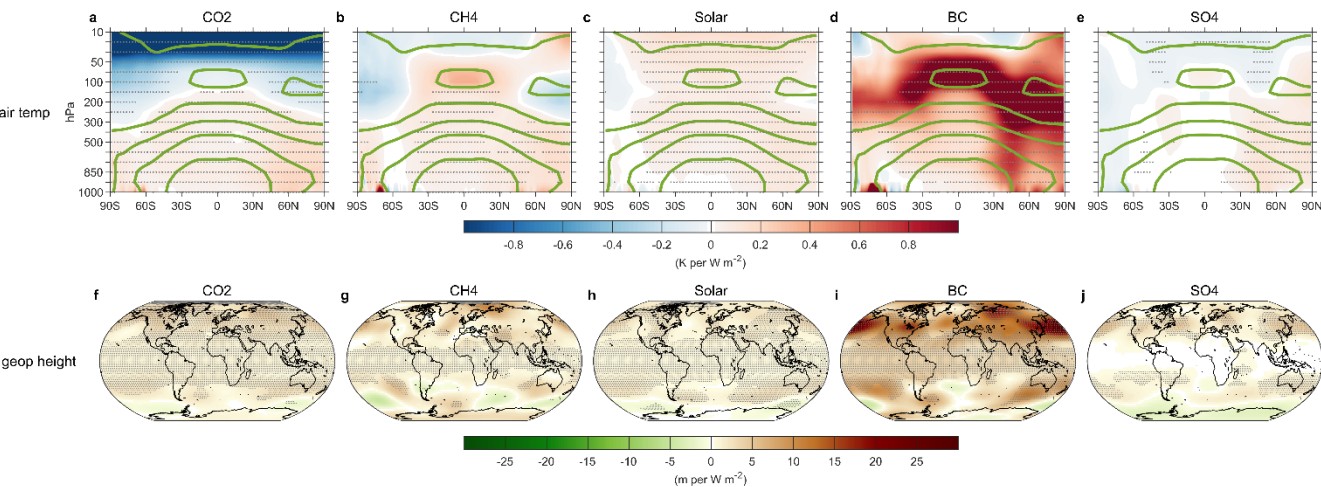

**Figure 7: MMM changes for zonal atmospheric temperature (a-e) and geopotential height of 500 hPa (f-j) per unit global TOA forcing. The thick green lines in the upper row are the climatology temperature in the control simulation. Grey dots indicate that the MMM changes are significant at a _p_ value of 0.05.**

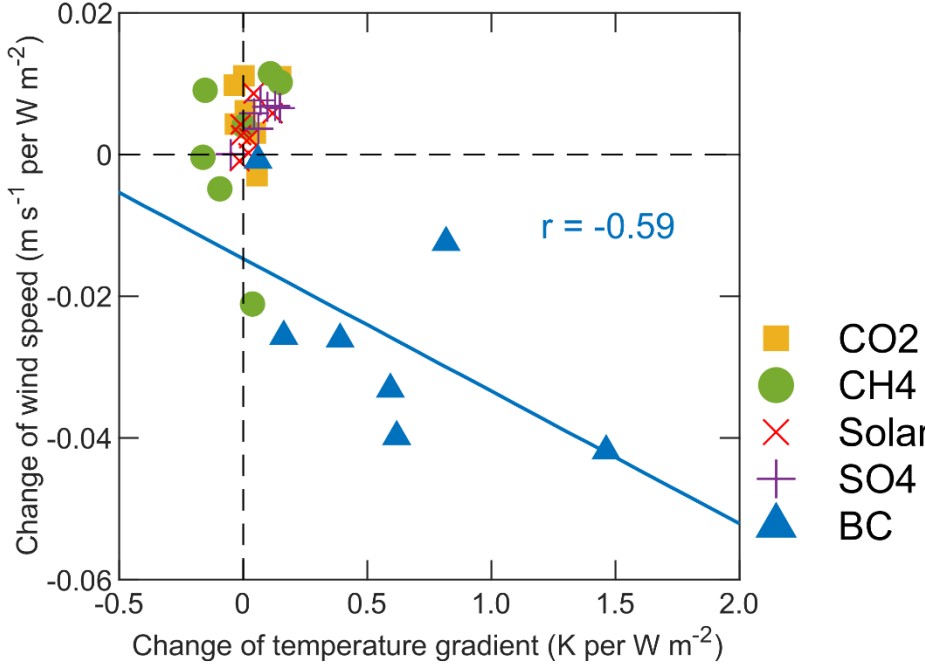

**Figure 8: Changes of wind speed versus changes of temperature gradient for each individual model and simulation. The CESM1-CAM4 model is excluded due to the unavailability of surface wind data. The linear correlation r is for the BC experiment.**