# Peer review of "Distinct surface response to black carbon aerosols"

_Atmospheric Chemistry and Physics, 2021_

## Author Comment (AC1)

**Response to comments #1**

Response: Thanks for your helpful and constructive comments. We have made several modifications and implemented the suggestions as needed. We describe a few major changes, followed by our response to individual comments.

i) We added ground heat term in the bar plot (Figure 4 and Figure S5);

ii) We added a table, summarizing global mean values of each energy budget term;

iii) We added the variable "humidity gradient" in Figure 6, as it may also influence latent heat flux, along with some discussions;

iv) We made some changes to global mean values in the previous version due to a coding mistake. These changes are tiny, and do not impact any conclusions reached.

General comment:
This study analyzed the PDRMIP multi-model ensemble to show that changes to the surface energy budget are distinctively different in the case of the BC-induced climate change from those enforced by other forcing agents such as CO2, scattering aerosol, and solar insolation. Specifically, the authors show that the climate response of the surface air temperature is governed by changes to multiple components of the surface energy budget in the BC case, contrary to other scenarios where surface radiative flux changes is the major factor that mostly determines the temperature response. I think this study is a nice follow up for recent studies that similarly assessed the surface energy budget change for some of the forcing agents to identify their different characteristics of climate responses. I would recommend the paper be considered for publication in ACP contingent upon if the authors appropriately address my specific concerns described below.

Response: thanks for your positive consideration and recommendation.

Specific comment:
Line 34-39: The numbers listed here should be shown with appropriate units (dimensions) since these numbers are for a unit BC forcing. For example, the energy flux changes (radiative, sensible and latent) should be dimensionless (rather than having the unit of $Wm^{-2}$) given that they are normalized by the TOA radiative flux change.

Response: corrected. We added 'per W $m^{-2}$' to all the units of changed variables in the abstract for clarity.

New abstract:

… … Globally, when a unit BC forcing was imposed at TOA, the net shortwave radiation at the surface decreased by -5.87±0.67 W $m^{-2}$ per W $m^{-2}$ (averaged over global land without Antarctica), which is partially offset by increased downward longwave radiation (2.32±0.38 W $m^{-2}$ per W $m^{-2}$) from the warmer atmosphere, causing a net decrease in the

incoming downward surface radiation of -3.56±0.60 W m$^{-2}$ per W m$^{-2}$. Despite a reduction in the downward radiation energy, the surface air temperature still increased by 0.25±0.08 K because of less efficient energy dissipation, manifested by reduced surface sensible (-2.88±0.43 W m$^{-2}$ per W m$^{-2}$) and latent heat flux (-1.54±0.27 W m$^{-2}$ per W m$^{-2}$), as well as a decrease of Bowen ratio (-0.20±0.07 per W m$^{-2}$). Such reductions of turbulent fluxes can be largely explained by enhanced air stability (0.07±0.02 K per W m$^{-2}$), measured as the difference of the potential temperature between 925 hPa and surface, and reduced surface wind speed (-0.05±0.01 m s$^{-1}$ per W m$^{-2}$). … …

Line 38-39: I'm a bit surprised that the sensible heat flux change (2.53Wm$^{-2}$) is larger than the latent heat flux change (1.30Wm$^{-2}$) simply because of my naïve understanding that the latent heat typically dominates the turbulent heat transfer from the surface to atmosphere. Can the authors explain why the opposite (i.e. sensible heat is larger than latent heat) occurs in the BC-forced scenario? The enhanced stability just explains the sum of latent and sensible heat changes but no explanation for their partitioning.

Response: yes, typically the latent heat flux is roughly four times as that of sensible heat. However, that is global mean values, combining both land and ocean. The large latent heat flux is mainly contributed by evaporation from ocean (~85%, e.g., *Schmitt, R. W., 2008: Salinity and the global water cycle. Oceanography, 21, 12–19.*). As we only focus on land grids in this study, the latent heat flux is greatly reduced. This is an interesting point, we added two sentences in the revised manuscript to note this.

We added the following text after Line 177:

"The larger change of H is somewhat contradicting to the common sense that λE dominates the turbulent flux on global mean scale. This is because on global scale, 85% of λE is from the ocean (Schmitt, 2008). In this study, we only focus on land grids, in which the λE is largely suppressed."

Section 2.2: I'm wondering if the energy balance described by equations (2) and (3) indeed applies when analysis is restricted to land grids only. Don't these equations need additional terms for energy exchange between land and ocean? Did the authors confirm that the balance relationship of (3) is indeed true in the model data analyzed? This can possibly be addressed by adding a bar for the residual in Fig. 4.

Response: accepted. Equations (2) and (3) have been widely used in the Earth science community, and to our best knowledge, no lateral heat exchange term between land and ocean could be found in any literature. We added the term ground heat (G, estimated as the residual) in the revised Figure 4, which shows that the term G is negligible in most cases. Thus, this term, if any, is presumably small and unlikely to be a large source of uncertainty in our analyses.

New Figure 4:

[Figure]

Minor point:
Line 101: relative -> relatively
Response: corrected.

Line 205: "wind speed (U) is likely the only main factor driving the change of latent heat flux": How about the specific humidity (qa)? The enhanced stability in the BC case can also change qa.

Response: added humidity gradient in the revised Fig. 6, along with some discussions. Please see the revised manuscript.

New figure 6

[Figure]

We added the following text after Line 228:

"Figure 6k-o show the changes of humidity gradient ($q_s$-$q_a$), defined as the specific humidity difference between the surface and 850 hPa. For $CH_4$, solar and $SO_4$, the gradient increases globally with values of 0.02 g kg$^{-1}$ per W m$^2$, 0.01 g kg$^{-1}$ per W m$^2$ and 0.01 g kg$^{-1}$ per W m$^2$ respectively, causing an increase of $\lambda E$ (Figure 3 and 4). For $CO_2$, the gradient shows slightly negative values (-0.002 g kg$^{-1}$ per W m$^2$), corresponding to reduced $\lambda E$ (Figure 3 and 4). In terms of BC, the humidity gradient increases by 0.06±0.02 g kg$^{-1}$ per W m$^2$, but with reduced $\lambda E$ flux, indicating that humidity gradient is not the primary driver of latent heat change. These analyses illustrate that humidity gradient may also influence latent heat flux for $CO_2$, $CH_4$, solar and scattering aerosols. For BC, on the other hand, change of wind speed should be the primary driver of the reduction of $\lambda E$ and humidity gradient is of less importance."

Line 256: I don't understand what 'bottom-up' means here. To my understanding, the energy balance constraint discussed throughout this paper is all 'top-down' regardless whether it is for top-of-atmosphere or surface. Why is it called 'bottom-up' when the surface energy response to solar insolation change is discussed? Please clarify.

Response: 'bottom-up' is a term widely used in the solar forcing study, in which the solar radiation directly heats the surface. The temperature responds firstly and then the impacts propagate upward. Our initial intention was to highlight that the influence of BC is somewhat similar to the solar forcing, as both of them have 'top-down' impact, in which BC and solar could modify the conditions of atmosphere first and then the impacts propagate downward to influence the surface.

We deleted these two terms to avoid confusion. In the revised version, we just say that the impact starts from the atmosphere and propagate downward to the surface (please see revised manuscript).

Revised abstract:

…… "These rapid adjustments under BC forcing occur in the lower atmosphere and propagate downward to influence the surface energy redistribution and thus, surface temperature response, which is not observed under greenhouse gases or scattering aerosols." ……

---

## Author Comment (AC2)

**Response to comments #2**

Response: Thanks for your helpful and constructive comments. We have made several modifications and implemented the suggestions as needed. We describe a few major changes, followed by our response to individual comments.

i) We added ground heat term in the bar plot (Figure 4 and Figure S5);

ii) We added a table, summarizing global mean values of each energy budget term;

iii) We added the variable "humidity gradient" in Figure 6, as it may also influences latent heat flux, along with some discussions;

iv) We made some changes to global mean values in the previous version due to a coding mistake. However, these changes are tiny, and do not impact any conclusion.

This paper explores the climate response at the surface to BC aerosol forcing in a set of sensitivity tests with a variety of global climate models. The paper draws a contrast between the surface response to BC forcing and the surface response to a variety of other climate forcing agents, including greenhouse gases, solar variations, and mostly scattering sulfate aerosols. The paper argues that the response at the surface to the attenuation of downward solar radiation attributable to BC aerosols elicits a significantly stronger compensating response in the surface turbulent fluxes compared to the other forcing agents. This is attributed to the increased static stability and closely related reduction in surface winds that the authors argue is more significant in response to BC aerosol forcing compared to the other forcing agents. The paper is useful contribution and the approach is fairly robust, apart from the usual limitations of global climate models. The paper is suitable for publication following some minor revisions.

Response: thanks for your positive consideration.

**Major comments:**
The individual figure elements in figures 2, 3, 5, 6 and 7 are very small. Presumably the purpose of showing a global map would be to reveal the regional differences evident in the map, only some of which are discussed in the text. Nevertheless, such variations are difficult to discern in maps as small as included here. If the intent is for the reader to simply zoom in on a computer screen, then I suppose it is sufficient. This is perhaps an editorial decision regarding whether the figures must be accessible to the reader in print form, or whether it is sufficient to presume the reader will zoom in electronically. Nevertheless, much of the argument rests on what are essentially global energy budget arguments with only limited discussion of the regional differences apparent across what amounts to more than 60 global maps. The authors could consider conveying more of the quantitative results in simpler bar figures, such as figure 4 and including global maps only for key results where the regional variations are central to the argument.

Response: accepted. Given the different responses of BC and different responses of source region and non-source region under BC, we still prefer to keep these figures, as

readers may be interested in the spatial maps. However, we included a Table summarizing the global mean values of the energy budget component in the revised version (Table 2) for the readers to look up.

Below is the Table 2 in the revised version:

Table 2. Globally-averaged multi-model mean (MMM±1s.e.) values of changes in surface energy components and temperature.

| Model | $\triangle$Rin (W m2 per W m2) | $\triangle\uparrow$LW (W m2 per W m2) | $\triangle$H (W m2 per W m2) | $\triangle\lambda$E (W m2 per W m2) | $\triangle$Bowen ratio (β per W m2) | $\triangle$G (W m2 per W m2) | $\triangle$T (K per W m2) |
|---|---|---|---|---|---|---|---|
| CO2 | 1.26±0.08 | 0.97±0.05 | 0.50±0.08 | -0.42±0.10 | -0.03±0.02 | 0.21±0.02 | 0.18±0.01 |
| CH4 | 1.02±0.11 | 0.68±0.06 | 0.01±0.05 | 0.14±0.02 | -0.09±0.04 | 0.19±0.04 | 0.12±0.01 |
| Solar | 1.11±0.03 | 0.47±0.04 | 0.19±0.03 | 0.26±0.03 | -0.05±0.02 | 0.20±0.01 | 0.08±0.01 |
| BC | -3.56±0.60 | 0.86±0.36 | -2.88±0.43 | -1.54±0.27 | -0.20±0.07 | 0.00±0.16 | 0.25±0.08 |
| SO4 | 1.54±0.14 | 0.54±0.09 | 0.32±0.06 | 0.44±0.07 | 0.00±0.02 | 0.24±0.01 | 0.10±0.02 |

The results shown in figures 2 and 3 describe the basic physics underlying the main point of the paper, but only for energy budget considerations over the land areas. Of course, from a global perspective, the total energy balance is strongly determined by the ocean response. The authors acknowledge that they are only focusing on land, but this choice is not justified in the discussion. Perhaps the notion is that the temperature response addressed in the following figures is more important to people over land because that is where the people are. If so, then the authors should state it, since the choice to ignore the ocean when considering surface energy balance responses to forcing agents is not intuitive.

Response: thanks. We added a sentence to note this in the Method section, where we first mention that our discussion is restricted to land grids only.

Below is the revised text since Line 111:

"In this study, we start from the surface energy balance. We restrict our discussions to land grids only, because this is where people live and thus, the temperature response over land is more important to the wellbeing of human."

The differences between the various forcing experiments shown in figure 5 look very dramatic. In particular, the BC experiment stands out strongly in comparison with the other forcing agents, which of course is part of the point of the paper. However, how much of that is simply because the surface forcing per unit of TOA forcing is greater for BC? Put another way, if the solar forcing were reduced by enough that the reduction in solar insolation at the surface were comparable to the attenuation of 10x BC, would there be important responses in the turbulent fluxes that are not evident in figure 5? Typically,

forcing from reductions in solar forcing, or forcing from purely scattering aerosols are of a similar magnitude at the surface as they are at the top-of-atmosphere. However, the surface forcing of a perturbation in BC at the surface can be several times that of the TOA forcing (see e.g. Magi et al. 2008 where the surface forcing efficiency of smoke can be approximately 10x the forcing efficiency at top-of-atmosphere).

Response: Excellent point. Yes, it is likely that the turbulent flux of solar forcing will be strongly enhanced if its surface forcing is similar to BC. We think it is an interesting feature that the surface forcing of BC is 10x larger than TOA forcing, which further demonstrates the importance to study surface forcing. We added this reference and noted this in our discussion.

We added some discussions after Line 184:

"It is also noted that the stronger responses in the BC scenario (Fig. 3, 4 and Table 2) could be partially related to its larger changes of surface radiative heating ($\triangle$Rin) compared with other forcing agents. Taking $CO_2$ as an example, $\triangle$Rin is 1.26 W m$^{-2}$, similar to its TOA forcing (1 W m$^{-2}$), whereas for BC, $\triangle$Rin is roughly three times larger (Table 2). Observations show that the surface forcing BC could be 10 times larger than TOA forcing on regional scales (Magi et al., 2008), indicating that BC could cause stronger changes of surface forcing than TOA forcing relative to other forcing agents."

In the abstract and discussion, the authors refer to a "top-down" influence of BC aerosols. I think I get what the authors are trying to imply by "top-down", but I find the explanation to be rather vague. For example, static stability is typically quantified by a potential temperature gradient in the lower troposphere. That is above the surface, I suppose, but not the top of the atmosphere, or even top of the troposphere. I think that if the authors are going to include this notion, especially in the abstract, they need to be quite a bit more specific about what exactly is defining the volume or geometry of the space they consider having a "top" and "bottom" and why a term that implies a downward action is appropriate. In my opinion it does not make the physical argument any clearer.

Response: accepted. We deleted these two terms for BC to avoid confusion in both abstract and discussion.

Revised abstract:

…… "These rapid adjustments under BC forcing occur in the lower atmosphere and propagate downward to influence the surface energy redistribution and thus, surface temperature response, which is not observed under greenhouse gases or scattering aerosols." ……

Finally, while I find that the robust results from an ensemble of simulations comparing responses to a range of individual forcing factors to be a valuable contribution to the literature, I find the notion that "a clear and detailed mechanism of surface radiative

response to BC is still lacking" (line 76) to perhaps be overstating the level of ignorance. While not purely a study of BC aerosol forcing, I would refer the authors to Liepert et al. (2004) for an early introduction to the notion that aerosol forcing at the surface can yield turbulent flux responses that are important on global scales.

Response: thanks for the reference. We added it in the introduction section. We also deleted that sentence and just say:

"The published studies citied above provide informative insights to the surface radiative responses to BC aerosols, but our understanding is still incomplete especially from the perspective of the surface energy balance."

**Minor comments:**
All of the quantitative forcing values noted in the abstract have positive signs, even while most of them are meant to quantify reductions in the forcing. Typically, reductions in a forcing value should be given a negative sign. Alternatively, and perhaps more rigorously, since many of the forcing quantities are directional (upward or downward) the authors could define changes that contribute an increase in the net downward forcing as positive in sign and decreases in net downward forcing as negative.

Response: corrected. We added negative signs to the negative values throughout the whole manuscript and mention this in the method section.

New abstract:

… … Globally, when a unit BC forcing was imposed at TOA, the net shortwave radiation at the surface decreased by $-5.87\pm0.67$ W m$^{-2}$ per W m$^{-2}$ (averaged over global land without Antarctica), which is partially offset by increased downward longwave radiation ($2.32\pm0.38$ W m$^{-2}$ per W m$^{-2}$) from the warmer atmosphere, causing a net decrease in the incoming downward surface radiation of $-3.56\pm0.60$ W m$^{-2}$ per W m$^{-2}$. Despite a reduction in the downward radiation energy, the surface air temperature still increased by $0.25\pm0.08$ K because of less efficient energy dissipation, manifested by reduced surface sensible ($-2.88\pm0.43$ W m$^{-2}$ per W m$^{-2}$) and latent heat flux ($-1.54\pm0.27$ W m$^{-2}$ per W m$^{-2}$), as well as a decrease of Bowen ratio ($-0.20\pm0.07$ per W m$^{-2}$). Such reductions of turbulent fluxes can be largely explained by enhanced air stability ($0.07\pm0.02$ K per W m$^{-2}$), measured as the difference of the potential temperature between 925 hPa and surface, and reduced surface wind speed ($-0.05\pm0.01$ m s$^{-1}$ per W m$^{-2}$). … …

In line 181 I think "and all grid were given" should read "and all grid cells were given".

Response: corrected.

Reference:

Magi, B.I., Fu, Q., Redemann, J. and Schmid, B., 2008. Using aircraft measurements to estimate the magnitude and uncertainty of the shortwave direct radiative forcing of

southern African biomass burning aerosol. Journal of Geophysical Research: Atmospheres, 113(D5).

Liepert, B.G., Feichter, J., Lohmann, U. and Roeckner, E., 2004. Can aerosols spin down the water cycle in a warmer and moister world?. Geophysical Research Letters, 31(6).